# BRAIN2MUSIC: RECONSTRUCTING MUSIC FROM HUMAN BRAIN ACTIVITY

## ABSTRACT

The process of reconstructing experiences from human brain activity offers a unique lens into how the brain interprets and represents the world. In this paper, we introduce a method for reconstructing music from brain activity, captured using functional magnetic resonance imaging (fMRI). Our approach uses either music retrieval or the MusicLM music generation model conditioned on embeddings derived from fMRI data. The generated music resembles the musical stimuli that human subjects experienced, with respect to semantic properties like genre, instrumentation, and mood. We investigate the relationship between different components of MusicLM and brain activity through a voxel-wise encoding modeling analysis. Furthermore, we discuss which brain regions represent information derived from purely textual descriptions of music stimuli. We provide supplementary material including examples of the reconstructed music at f2mu.github.io

## 1 INTRODUCTION

Music holds universal significance, acting as a medium for expression and communication across diverse cultures. The representation of music within our brains has been a significant topic of interest in neuroscience. Previous studies have examined human brain activity, as captured by functional magnetic resonance imaging (fMRI), while participants listened to music. These studies discovered musical feature representations in the brain, such as rhythms (Alluri et al., 2012), timbres (Toiviainen et al., 2014; Allen et al., 2018), emotions (Koelsch et al., 2006), and musical genres (Casey, 2017; Nakai et al., 2021). This body of research provides valuable insights into how music-related features – both objective and subjective, acoustic and semantic – are represented within the brain, illuminating the complexity of our experiences with music.

With the advent of text-to-music models, conditional generation of high-fidelity music has become feasible. This exciting development bridges the gap between our linguistic understanding of music and the actual creation of musical compositions. Thus, new questions arise: How do the text and music embeddings used by these music generation models correspond to representations in the human brain? Furthermore, if a correspondence exists, is it possible to generate music directly from brain activity?

In this paper, we explore the feasibility of reconstructing music from brain activity scans with MusicLM (Agostinelli et al., 2023). We also compare some of the internal representations of MusicLM, such as the high-level semantic embedding space of MuLan (Huang et al., 2022) and the low-level, averaged embeddings from w2v-BERT, to activity in different brain regions, providing novel insights.

## 2 RELATED WORK

**Music generation models.** Generating high-fidelity music has been challenging due to the need to produce music with both high-quality audio and long-term consistency. An initial approach was introduced by Jukebox (Dhariwal et al., 2020), which proposes a hierarchical structure at different time resolutions, modeled autoregressively with Transformer-based models. While Jukebox generates music with high temporal coherence, it contains perceptible artifacts. PerceiverAR (Hawthorne et al., 2022) makes use of SoundStream, a neural audio codec which compresses audio at low bitrates, while

maintaining a high-quality reconstruction (Zeghidour et al., 2021a). PerceiverAR models a sequence of discrete SoundStream embeddings autoregressively, generating high quality audio but lacking in temporal coherence.

Recently, auto-regressive and diffusion-based models have significantly advanced the quality of synthesis in both music and broader audio generation. AudioLM (Borsos et al., 2022) suggests autoregressively modeling a hierarchical tokenization scheme composed of both *semantic* and *acoustic* discrete audio representations. MusicLM (Agostinelli et al., 2023) integrates the AudioLM framework with the joint music/text embedding model MuLan (Huang et al., 2022), enabling the generation of high-fidelity music conditioned on detailed text descriptions. Other Transformer-based methods encompass Donahue et al. (2023), Lam et al. (2023a), and Copet et al. (2023), with the last of these leveraging the EnCodec audio codec (Défossez et al., 2022). Additionally, diffusion model-based strategies for music generation are used by Riffusion (Forsgren & Martiros, 2022), with the most recent advancements proposed in (Huang et al., 2023; Liu et al., 2023; Ghosal et al., 2023; Lam et al., 2023b).

Within the framework of our Brain2Music pipeline, we employ MusicLM and its constituent components. However, our methodology is fundamentally adaptable to any music generator, provided the generator can accommodate conditioning on a dense embedding. When this project began, MusicLM was the only potent music generation model accessible to us.

**fMRI audio decoding and fMRI encoding.** One of the key goals in neuroscience is to understand the representations that govern the relationship between brain activity and sensory and cognitive experiences. To this end, researchers construct encoding models to quantitatively describe which features of these experiences (e.g., color, motion, and phonemes) are encoded as brain activity. In contrast, they also build decoding models to infer the experienced content from a specific pattern of brain activity (for a review, see Naselaris et al. (2011)).

Particularly in recent years, researchers have discovered correspondences between the internal representations of deep learning models and those of the brain across various sensory and cognitive modalities (Yamins et al., 2014; Kell et al., 2018). These findings have advanced our understanding of brain functions through (a) the development of encoding models mediated by the representations (Güçlü & van Gerven, 2015), (b) interpretations of the representations based on their correspondence with brain functions (Cross et al., 2021; Takagi & Nishimoto, 2022), and (c) reconstruction of experienced content (such as visual images) from brain activity (Shen et al., 2019; Chen et al., 2023; Takagi & Nishimoto, 2023). More specifically, in the context of investigating auditory brain functions, researchers have developed encoding models using deep learning models that process auditory inputs (Kell et al., 2018), and conducted studies to reconstruct perceived sounds from brain activity (Santoro et al., 2017; Park et al., 2023). However, so far, these studies have largely targeted general sounds, including voices and natural sounds. There are no instances of constructing encoding models using the internal representations of text-to-music generative models, or reconstructing musical experiences from brain activity with a focus on music and its unique features.

## 3 METHODS

### 3.1 MUSIC FMRI DATASET

We pre-process the *music genre neuroimaging dataset*[1] from Nakai et al. (2022) in a same manner as Nakai et al. (2021). We outline details of the collection and preprocessing protocol in Section A.1.1. The dataset contains music stimuli from 10 genres (blues, classical, country, disco, hip-hop, jazz, metal, pop, reggae, and rock) which were sampled randomly from the (music-only) dataset GTZAN (Tzanetakis & Cook, 2002). A total of 54 music pieces (30s, 22.050kHz) were selected from each genre, providing 540 music pieces. A 15s music clip was selected at random from each music piece. The dataset contains 480 examples for training and 60 for reporting the final results.

In this work, we augment the original dataset (Nakai et al., 2022) by introducing English text captions which we have made publicly available[2]. These captions, averaging approximately 46 words or 280

---

[1]Download link: openneuro.org/datasets/ds003720
[2]Download link: f2mu.github.io/caption-dataset.csv

characters in length, typically describe the musical pieces in terms of genre, instrumentation, rhythm, and mood. They often comprise fragmented or semi-complete sentences, with an average of about 4.5 sentences per caption. The style of writing is subjective, reflecting not only the technical components of the music such as the instruments used, but also the emotional responses or atmospheres they might evoke in listeners. Captions were written in Japanese and translated using DeepL. Several exemplar captions and the instructions given to the raters are in Section A.4. In this paper, the captions are used to study how purely semantic, text-derived embeddings relate to brain activity induced by the corresponding music stimuli.

## 3.2 MuLan and MusicLM

MuLan (Huang et al., 2022) is a joint text/music embedding model consisting of two towers, one for text (MuLan$^{\text{text}}$) and one for music (MuLan$^{\text{music}}$). The text tower is a BERT (Devlin et al., 2019) model pre-trained on a large text corpus. For the audio tower we use the ResNet-50 (He et al., 2015) variant. MuLan's training objective is to minimize a contrastive loss between the 128-dimensional embeddings produced by each tower for an example pair of aligned music and text. For example, the embedding of a rock song's waveform is supposed to be close to the embedding of the text *rock music* and far from *calming violin solo*. In this paper, if we refer to a MuLan embedding we mean by default the embedding of the music tower.

MusicLM (Agostinelli et al., 2023) is a conditional music generation model. Conditioning signals include – but are not limited to – text, other music, and melody. In our decoding pipeline MusicLM is conditioned on a MuLan embedding which we compute based on an fMRI response. In MusicLM, music is generated in two consecutive stages. The first stage learns to map a MuLan embedding to a sequence of w2v-BERT tokens. These tokens used in MusicLM are extracted from a w2v-BERT (Chung et al., 2021) model's activations in the 7th layer, by clustering them with k-means. MusicLM's second stage maps the w2v-BERT tokens from the first stage and the MuLan embedding to acoustic tokens. These stem from a SoundStream (Zeghidour et al., 2021a) model's residual vector quantizer. The resulting tokens are converted back into audio using a SoundStream decoder. As in AudioLM (Borsos et al., 2022), the second stage is split into a coarse and fine modeling stage. All three stages are implemented as Transformer models. A visualization of MusicLM's components is in Figure A.1.

## 3.3 Decoding: Reconstructing Music from fMRI

With *decoding* we refer to attempting the reconstruction of the original stimuli (to which a test subject was exposed) based on their recorded brain activity. This process can be subdivided into (1) predicting the music embedding based on fMRI data and (2) retrieving or generating music based on that embedding.

**Music embedding prediction from fMRI data.** Let $\mathbf{R} \in \mathbb{R}^{n \times s \times d_{\text{fmri}}}$ denote the response tensor (obtained via fMRI for each of the five participants), where $n = 540$ is the number of stimuli (i.e., 15s music clips), $s = 10$ is the number of fMRI scans per stimulus (15s), and $d_{\text{fmri}}$ is the number of voxels. $d_{\text{fmri}}$ varies slightly across subjects depending on the physical size of their brain. For subject 1 it is around 60k. Our prediction targets are the music embeddings of the stimulus (e.g., MuLan), $\mathbf{T} \in \mathbb{R}^{n \times r \times d_{\text{emb}}}$, with $r$ being the number of embeddings computed per 15s clip (which depends on the embedding model's window size and the constant step size of 1.5s by which we advance this window). Table A.2 lists the embeddings, derived from models present in MusicLM, that we consider as candidate *music embeddings* in the Brain2Music architecture.

To align $\mathbf{R}$ and $\mathbf{T}$ along the time dimension, we average entries in $\mathbf{R}$ in a sliding-window fashion to match the time ranges for which feature vectors in $\mathbf{T}$ were computed. For example, to predict the MuLan embedding ranging from 0s to 10s (due to MuLan's window size being 10s) we rely on the average of five fMRI scans (from 0-1.5s, 1.5-3s, ..., 9s-10.5). This leaves us with $m := n \times r$ pairs of response and target, which we split following Nakai et al. (2022). We model the relationship between music embeddings and responses with weight matrix $\mathbf{W} \in \mathbb{R}^{d_{\text{fmri}} \times d_{\text{emb}}}$:

$$\hat{\mathbf{T}} = \mathbf{R}\mathbf{W} \, . \tag{1}$$

We use an L2-regularized linear regression to estimate $\mathbf{W}$ on the training dataset. Note that there is no generalization between different subjects, because of anatomical differences. For each subject and

target dimension, the regression regularization hyperparameter is tuned with five-fold cross validation on the training dataset, while a test split is held out for later evaluation. Details are in Section A.1.3.

The training is independently performed for each anatomically defined region of interest (ROI) from the Destrieux atlas (Destrieux et al., 2010). An ROI is a group of voxels. From all 150 ROIs we select the top $n_{\text{ROIs}} = 6$ with the highest correlation scores (as determined via cross-validation) and create an ensemble model by averaging their predictions. Concretely that leaves us with ROIs varying in size[3]. The median across all subjects is 138.5 voxels; the average is 258.6 voxels. Although the exact location of the ROIs chosen for each subject vary, for all subjects, ROIs are chosen primarily from the auditory cortex. For a given 15s music stimulus we predict $r$ many embeddings (depending on the chosen embedding type).

**Music retrieval and music reconstruction.** We explore two different approaches to derive the original stimulus from the prediction $\hat{\mathbf{T}}$ namely retrieving similar music from an existing music corpus and generating music with MusicLM.

For the retrieval we compute the MuLan embeddings for the first 15s of every music clip in the Free Music Archive (FMA) (Defferrard et al., 2017). Unless stated otherwise, we use the *large* variant which contains a wide range of diverse music; concretely, 106,574 music tracks from 161 unbalanced genres. The *retrieved music* is the audio of the clip whose embeddings are the closest to the predicted one as measured by the cosine similarity.

Alternatively, we can *generate* the music by conditioning the MusicLM model (a high-level overview is in Section 3.2) on the predicted embeddings. The model can then be used to generate music conditionally. To condition MusicLM we average the $r = 4$ predicted MuLan embeddings along the time dimension. This is not strictly necessary, but we empirically found the generated outputs to be more stable compared to a version in which we provided all four embeddings to the model.

The two methods, retrieval and generation, have different advantages and disadvantages. The retrieval approach constrains the faithfulness by its limited size. The predicted embedding could potentially contain rich information about the song which is partially lost by mapping it onto its *nearest* neighbor in the dataset. The generative model, on the other hand, can in theory generate any kind of music covered by its training distribution, making it conceptually more powerful. That includes tracks which are not training examples (e.g., combinations of musical concepts). A disadvantage of this method is that the generation model may not adhere precisely to the provided embedding.

**Evaluation metrics.** Following the decoding literature (Takagi & Nishimoto, 2022; Park et al., 2023), we compute an *identification accuracy* of the predicted $d$-dimensional embeddings with respect to their target embeddings. Details are spelled out in Section A.1.4. As a second metric we also use *top-$n$ class agreement* based on the LEAF (Zeghidour et al., 2021b) classifier operating on AudioSet classes (Gemmeke et al., 2017). In this context, we compute the per-class probabilities for original and reconstructed music. We then look at three groups of music-related classes, namely genres, instruments, and moods. For each group we compute the top-$n$ agreement measuring how much overlap there is between the top-$n$ most probable class labels computed for original and reconstruction. The full list of AudioSet class names in each group is in Section A.1.6.

## 3.4 ENCODING: WHOLE-BRAIN VOXEL-WISE MODELING

To interpret the internal representations of MusicLM, we examine the correspondence between them and recorded brain activity. More specifically, we build whole-brain voxel-wise encoding models to predict fMRI signals using different music embeddings occurring in MusicLM: audio-derived embeddings (MuLan$^{\text{music}}$ and w2v-BERT-avg), and text-derived embeddings (MuLan$^{\text{text}}$).

We first build encoding models to predict voxel activity from the audio-derived embeddings: MuLan$^{\text{music}}$ and w2v-BERT-avg. Next, we build encoding models using audio-derived MuLan$^{\text{music}}$ and text-derived MuLan$^{\text{text}}$ embeddings to predict fMRI signals. This allows us to explore the differences between these two types of embeddings. The text-derived embeddings are particularly interesting to study, because they can – by definition – only represent the high-level information

---

[3]ROI sizes vary between subjects. The top six ROIs of subject 1, for example, which are the most predictive of MuLan embeddings, have the dimensionalities 61, 70, 109, 218, 296, and 706.

contained in the music caption they are computed from. The MuLan[text] embeddings we use have a 1:1 correspondence to GTZAN clips and are computed by inferring MuLan's text tower (a fine-tuned BERT model) for a given GTZAN clip's text caption. Caption examples are in Section A.4.

We also compared prediction performance of MuLan[music] with other audio-derived features for five different anatomically defined auditory subregions: A1 to A5 (Glasser et al., 2016). Melspectrogram was used as a low-level feature, while HuBERT (Hsu et al., 2021) and wav2vec2.0 (Baevski et al., 2020) were used as high-level features. A key difference between MuLan[music] and HuBERT/wav2vec2.0 is that MuLan[music] is trained on music, whereas the latter are trained on speech. Specific details of each embedding are in Section A.1.9. In addition, to determine whether MuLan[music] encompasses more than genre information, we compare the prediction performance of the MuLan[music] model versus one-hot vectors of music genre for each GTZAN clip.

The training data preparation is done in the same manner as in the decoding (outlined in Section 3.3). The modeling problem is inverse to Equation 1, i.e., fMRI responses are predicted based on different embeddings. Model weights are estimated from training data using L2-regularized linear regression, and subsequently applied to test data. We estimate weights of the model from training data, and regularization parameters are explored during the training using five-fold cross-validation. For evaluation, we use Pearson's correlation coefficients between predicted and measured fMRI signals. We compute statistical significance by comparing the estimated correlations to the null distribution of correlations calculated from the shuffled data. We set the threshold for statistical significance at $P < 0.05$, and corrected for multiple comparisons using the FDR procedure. For MuLan[text] and one-hot music genre vectors, we perform up-sampling to match MuLan[music]'s sampling rate.

## 4 RESULTS

### 4.1 DECODING (FMRI TO MUSIC)

**Music embedding prediction.**    Going from fMRI to music requires the prediction of an intermediate music representation, that is, selecting the music embedding to use. In our architecture, the choice of the music embedding represents a bottleneck for the subsequent music generation: Only if we can predict music embeddings close to music embeddings of the original stimulus heard by the subject, will we be able to generate music that is similar to the original stimulus with MusicLM. The reconstruction quality is further constrained by what the embedding can capture.

Results obtained when predicting different embedding types are reported in Table 1. We find that MuLan[music] embeddings can be more accurately predicted from fMRI signals than MuLan[text], w2v-BERT-avg, or SoundStream-avg embeddings. When predicting MuLan[music] embeddings, we observe the highest correlation as well as the highest identification accuracy. Three reasons may contribute to this: (1) The MuLan[text] and MuLan[music] embeddings, may be more closely aligned to the high level of abstraction represented in the fMRI data than the other embedding types. (2) The heuristics we applied to align w2v-BERT and SoundStream embeddings to the fMRI data may not be optimal and may require further analysis. (3) MuLan[music] embeddings have access to more musical information than MuLan[text] embeddings because they are computed from the audio signal. Based on this finding, in the remainder of this section, we use the fMRI data to predict MuLan[music] embeddings (for brevity referred to as MuLan embeddings) and use them to reconstruct the original stimulus.

**Quantitative reconstruction evaluation.**    Figure 1 shows the main quantitative results. We use two quantitative measures to evaluate the similarity of reconstructed music and original stimulus: identification accuracy (for two embeddings of different semantic level of abstraction) and AudioSet top-$n$ class agreement.

The quality limit imposed by retrieval from FMA is estimated via an *oracle* predictor. It simply bypasses the linear regression and instead retrieves an FMA clip based on the original music stimulus' MuLan embedding. It simulates the retrieval performance we would achieve if our fMRI to MuLan prediction was perfect. Performance achieved by a model sampling randomly from FMA is indicated by the *chance* result in the plots.

Overall we observe significant above-chance performance on all metrics, establishing strong support for our ability to extract musical information from the fMRI scans. The identification accuracy

Table 1: Comparison of different music embedding prediction targets (fMRI-to-embedding). The reported error is the standard deviation across five test subjects. Identification accuracy and correlation are computed between embeddings of GTZAN audio (computed with the embedding model listed in the respective row) and the embeddings predicted by the regression model. The magnitude of the correlation is strongly influenced by the expressiveness of the chosen embedding. The main takeaway from it is to establish an ordering of the different possible regression targets. To further analyze the behavior, we plot the predicted MuLan embeddings in Figure A.6 and observe a noisy clustering. An analysis of the regression regularization hyperparameter tuning, which is critical to avoid overfitting, can be found in Section A.1.3; more detailed properties of the embeddings in Table A.2.

| Embedding | Identification accuracy | | Correlation | |
|---|---|---|---|---|
| | Test | Train | Test | Train |
| SoundStream-avg | $0.674 \pm 0.016$ | $0.764 \pm 0.029$ | $0.184 \pm 0.020$ | $0.255 \pm 0.009$ |
| w2v-BERT-avg | $0.837 \pm 0.005$ | $0.941 \pm 0.007$ | $0.113 \pm 0.003$ | $0.167 \pm 0.006$ |
| MuLan[text] | $0.817 \pm 0.014$ | $0.877 \pm 0.009$ | $0.181 \pm 0.012$ | $0.245 \pm 0.009$ |
| MuLan[music] | $0.876 \pm 0.015$ | $0.992 \pm 0.003$ | $0.307 \pm 0.016$ | $0.538 \pm 0.023$ |

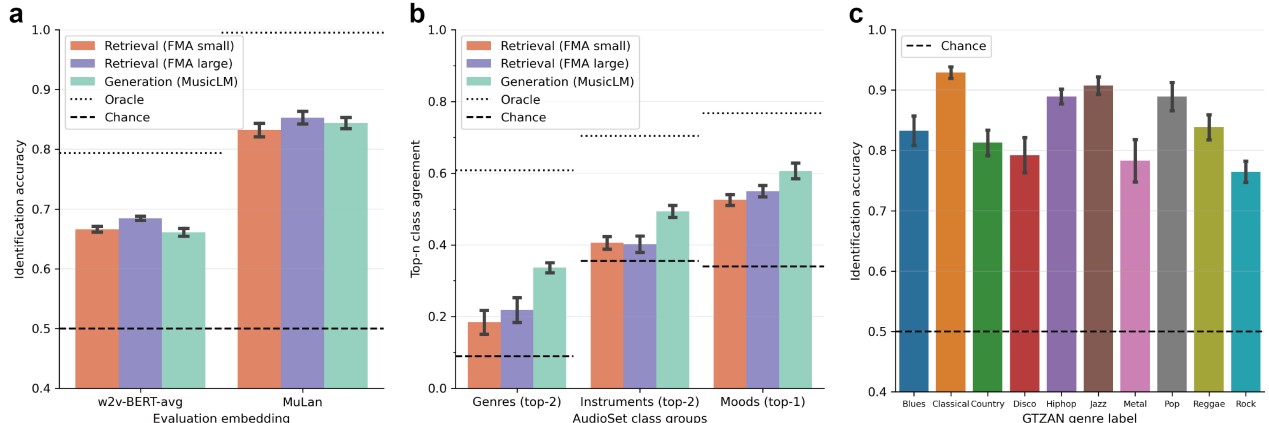

Figure 1: Main quantitative results of the decoding, i.e., music reconstruction. The dashed, horizontal lines (chance) indicate the performance of a random music predictor (sampling from the FMA dataset). The dotted lines (oracle) are the oracle performance, corresponding to the performance achieved by a regressor which would predict exactly the ground truth MuLan embedding of GTZAN. Error bars indicate standard error of the mean across five subjects. **a** Identification accuracy for different evaluation embeddings. Identification accuracy computed between the embeddings of the original stimulus (music) and the embeddings of the reconstructed music. The reconstructed music is more similar to the stimulus it was derived from with respect to high-level embeddings (MuLan) than the low-level w2v-BERT-avg. Differences between generation and retrieval on FMA large (about 106k clips) are marginal, whereas retrieving from FMA small (8k clips) is overall worse. **b** AudioSet top-$n$ class agreement for different groups of AudioSet classes. A list of the classes in each group is in Section A.1.6. Generation is here significantly superior to retrieval from FMA (both small and large). The worst performance – relative to chance and oracle – is attained on the instrument agreement. **c** Identification accuracy (based on MuLan embeddings of original and generated music) shown separately for each of the GTZAN genres. The model performance is consistent across all genres.

comparison across different embedding types hints at our reconstruction to be most faithful to the original stimulus with respect to high-level semantic features as captured by MuLan. While this might seem unsurprising, given MuLan is the target embedding of our prediction, it is not necessarily granted that high-level semantic information about perceived music is contained in the recorded brain response in the first place.

Whether or not low-level information is contained in the reconstruction is measured by the identification accuracy on w2v-BERT-avg. Note, however, that the results are confounded by the embedding

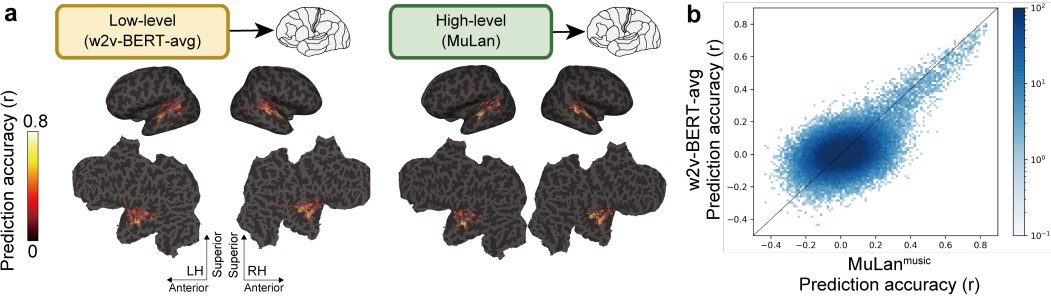

Figure 2: **a** Prediction performance (measured using Pearson's R) for the voxel-wise encoding model applied to held-out test music on subject 1, projected onto the inflated (top, lateral and medial views) and flattened cortical surface (bottom, occipital areas are at the center), for both left and right hemispheres. Brain regions with significant accuracy are colored (all colored voxels $P < 0.05$, FDR corrected). **b** Density plot of the MuLan$^{music}$ (x-axis) versus w2v-BERT-avg (y-axis) model prediction accuracy. Darker colors indicate a higher number of voxels in the corresponding bin.

averaging we perform for w2v-BERT to align the temporal resolution of the embeddings with that of the fMRI scans. However, the numbers are in-line with the qualitative observation we make, that is, low-level, acoustic features are relatively not well aligned, whereas the overall music style is. Our qualitative analysis is in Section A.2.

We further observe consistent prediction performance across different genres (as labeled in the GTZAN dataset). The highest accuracy is achieved on classical music, which is likely due to its distinctive musical style.

## 4.2 ENCODING (BRAIN ACTIVITY PREDICTION)

**Comparison between different audio-derived embeddings.**
Figure 2a shows the prediction accuracy of the encoding models for different types of audio-derived embeddings of music within MusicLM: MuLan and w2v-BERT-avg. MuLan embeddings tend to have higher prediction performance in the lateral prefrontal cortex than w2v-BERT-avg, suggesting that MuLan captures high-level music information processed in the human brain. However, when we focus on the auditory cortex, both of the embeddings have some degree of correspondence with human brain activity in the auditory cortex. Given that the text-music model used in this study was not brain-inspired compared to the previous deep learning model such as convolutional neural network, it is intriguing that this correspondence with the brain emerged. In addition, although each embedding represents different levels of audio-derived embeddings from low (w2v-BERT-avg) to high (MuLan), they predicted fairly similar brain regions within the auditory cortex. Figure 2b further confirms that well predicted voxels are largely overlapping between two embeddings.

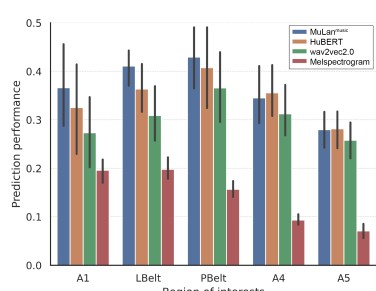

Figure 3: Prediction performance by different models for different ROIs within auditory cortex. Error bars indicate standard error of the mean across subjects.

These results suggest that, unlike the visual cortex (Takagi & Nishimoto, 2022), there is not as strong of a hierarchical functional differentiation of audio-derived embeddings in the auditory cortex as previously thought. Note, as we mentioned at 4.1, that the results are confounded by the embedding averaging we perform for w2v-BERT. Please see also the limitations in Section 5. We provide additional results for all subjects in Figure A.8.

Figure 3 shows MuLan$^{music}$ outperformed the other models overall. As expected, the prediction accuracy of melspectrogram decreased with the increasing hierarchy of the auditory cortex. However, strikingly, DNN features more accurately predicted Lbelt/Pbelt regions (also known as A2/A3) than A1, yet demonstrated lower performance in A4/A5. This trend has not been observed in other auditory studies, suggesting it might be unique to music.

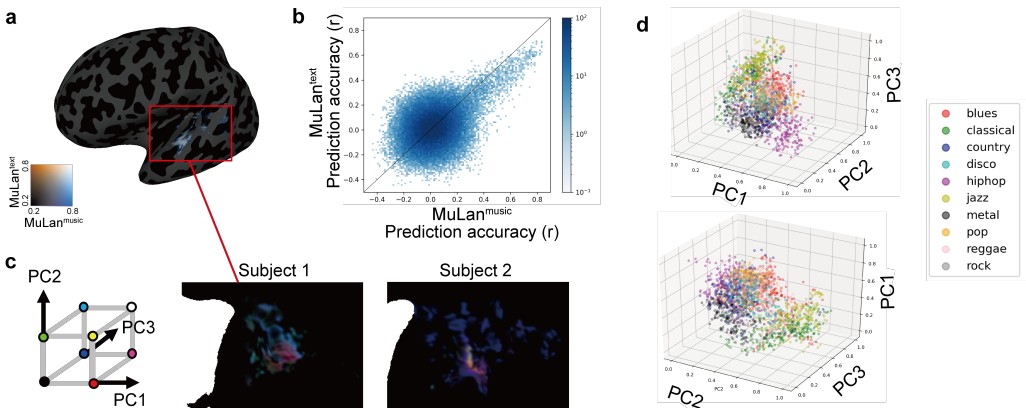

Figure 4: **a** Prediction performance on subject 1 for MuLan$^{text}$ model versus MuLan$^{music}$ model. All colored voxels $P < 0.05$, FDR corrected. The area in the red rectangle corresponds to the auditory cortex. **b** Density plot of the MuLan$^{music}$ (x-axis) versus MuLan$^{text}$ (y-axis) model prediction accuracy. **c** Principal components analysis reveals gradient from medial to lateral axis within auditory cortex. **d** Visualisation of the representational relationships between each music projected onto the PC space. The colors represent the genre to which each sample belongs.

**Comparison between audio- and text-derived MuLan embeddings.** We next investigate how much text-derived, abstract information about music is differently represented in the auditory cortex compared to audio-derived embeddings.

Figure 4a shows the prediction performance of the encoding models for MuLan$^{text}$ versus MuLan$^{music}$. We provide additional results for all subjects in Figure A.8. It shows that for some subjects, the inner side of the sulcus represents music stronger than outer side. However, there still seems to be modest functional differentiation in the brain. Although these two representations are trained to match (Huang et al., 2022), due to the many-to-many nature of text and music pairings the objective cannot be achieved perfectly. We show that, from a neuroscience perspective, MuLan$^{music}$ and MuLan$^{text}$ actually acquired fairly similar representations. Figure 4b further confirms that well-predicted voxels are largely overlapping between two embeddings. Finally, we investigated unique contributions of MuLan$^{music}$ and MuLan$^{text}$ and confirmed that MuLan$^{text}$ lacks unique explained variance power when compared to MuLan$^{music}$ (see A.1.8 for the detail).

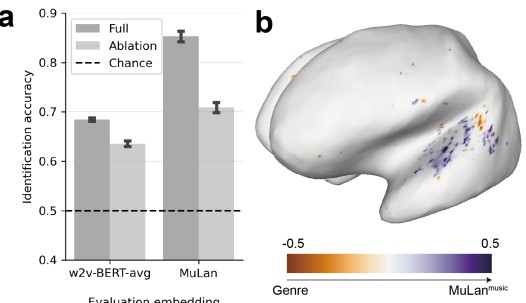

Figure 5: **a** Comparison of identification accuracy between the model in Figure 1 (full) and those models trained with one-genre-out ablation data and tested on the ablated genre (ablation). **b** Comparing prediction performance obtained by MuLan$^{music}$ and genre model on subject 1. Only the voxels with $R > 0.4$ are colored.

To further examine information representation in representative voxels, we perform principal component analysis (PCA) on the weight matrix of encoding model of MuLan$^{music}$ (Figure 4c; see A.1.7 for details and Figure A.8 for all subjects results). The top three PC components explained 25.8, 11.0 and 5.4% of the total variance, respectively. We observed a gradient from PC1 to PC2 or PC3 from the medial to lateral axis within auditory cortex, which is not apparent from comparisons between MuLan$^{music}$ and w2v-BERT. While we can observe this gradient consistently across subjects, there is also notable individual differences regarding the distribution of PC2 and PC3. To interpret the PCs, we also project each music onto a PC axes (Figure 4d). The music projected onto the three-dimensional PC space is rather intricately distributed, although there are some coarse clusters according to genres (e.g. Hiphop and Classical are at opposite ends of the spectrum).

### 4.3 Generalization Beyond Music Genre

We next investigate whether our model can generalize to the music genre that was not used during training. To do so, we ablate one genre during training and determine the identification accuracy on clips of the held-out genre (from the test set). Figure 5a shows that our model performs significantly above chance on the unseen music genres. This suggests that our reconstruction method is generalizing beyond the genres present in the training data.

We further compare the prediction performances of the MuLan$^{music}$ and genre models to test whether our encoding model captures information beyond music genres. We find that, compared to the music genres vectors, the MuLan$^{music}$ embeddings predict activity in the auditory cortex more broadly and with higher accuracy (Figure 5b). This is another piece of evidence that our model might predict beyond the music genre information. We provide additional results for all subjects in Figure A.9.

## 5 Limitations

The powerful music generation model we use converts a MuLan embedding into music. Information that is not contained in the embedding, but required to produce a music clip, is added by the model. In the retrieval case, it is even guaranteed that the reconstruction is musical, because it is directly pulled from a dataset of music. While this leads to impressive, human-digestible results, it also suggests a higher level of reconstruction quality than there may be.

The main three factors limiting the reconstruction quality are: (1) how much information we can extract with linear regression from the fMRI data, (2) what the chosen music embedding – in our case MuLan – can capture, and (3) the limitations of the music retrieval (dataset size and diversity) or generation (generative model capabilities). In the encoding analysis in Section 4.2 we investigate the limitations of (1 + 2) together by predicting brain activity for different embedding types. We disentangle (2 + 3) from (1) by showing the maximum attainable performance as an oracle dashed-line in Figure 1. It shows, for example, that top-1 class agreement on moods cannot exceed 78% given the MuLan embedding and FMA retrieval dataset choice. To observe the variability of (3), we experiment with three different reconstruction processes, i.e., two sizes of FMA for retrieval and a generative model. Further investigation of (1), (2), and (3) remains an open challenge.

The coarse temporal sampling rate of fMRI (1.5s, in the present study) is a limitation of the present study. However, it is noteworthy that even at the fMRI sampling rate of 2.5s, Santoro et al. (2017) showed temporal specificity at 200ms by using multi-voxel patterns. Different voxels might have information about different frequency bands, which may collectively contribute to this result. How much information can actually be retrieved from fMRI is a subject for future research (see Nishimoto et al. (2011) for reconstructing perceived natural movies from fMRI data using frequency-decomposed voxel-wise representations). Similarly, the temporal averaging of SoundStream and w2v-BERT embeddings likely worsens their expressiveness.

## 6 Conclusion

With Brain2Music we explored the exciting research direction of reconstructing music from recorded human brain activity. By conditioning MusicLM on a dense music embedding predicted from fMRI data, we were able to generate music which resembles the original music stimuli on a semantic level. We also investigated the connection between a text-to-music model and the human brain in a quantitative manner by constructing an encoding model. Specifically, we assessed where and to what extent high-level semantic information and low-level acoustic features of music are represented in the human brain. Although text-to-music models are rapidly developing, their internal processes are still poorly understood. This study is the first to provide a quantitative interpretation from a biological perspective.

Given the nascent stage of music generation models, this work is just a first step. Future work may improve the temporal alignment between reconstruction and stimulus or explore the reconstruction of music from pure imagination.

## ETHICS STATEMENT

Our work addresses the challenging task of reconstructing music from human brain activity. This is exploratory in nature and has the sole goal of better understanding the human brain and how it can be decoded. Accessing sensitive cognitive or emotional information is an ethically critical field. Participants of the data collection, which was carried out by Nakai et al. (2022), have explicitly consented to the data collection for research purposes. Experiments that contributed to this work were approved by an Institutional Review Board (IRB).

Note that the decoding technology described in this paper is unlikely to become practical in the near future. In particular, reading fMRI signals from the brain requires a volunteer to spend many hours in a large MRI scanner. While there are no immediate privacy implications from the technology as described here, as with this study, any such analysis must only be performed with the informed consent of the studied individuals. With the amount of work required to obtain fMRI signals from the brain, we do not have direct applications in products in mind.

## REPRODUCIBILITY STATEMENT

We encourage researchers to reproduce our work and support them in several ways. The main dataset we use is open-source Nakai et al. (2022). It contains pairs of music and fMRI scans and is at the heart of what we explore. The additional caption data which we have collected is being shared as part of this work[4]. The weights of the music generation model MusicLM and the joint text+music embedding model MuLan are proprietary. However, we publish the source code used to run the experiments predicting music embeddings from fMRI, which also contains the evaluation implementation.

---

[4]Download link: f2mu.github.io/caption-dataset.csv

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

# A APPENDIX

## A.1 METHOD DETAILS

### A.1.1 FMRI DATA COLLECTION AND PREPROCESSING

**Data collection details.** During scanning, five participants were asked to focus on a fixation cross at the center of the screen and to listen to the music clips through MRI-compatible insert earphones (Model S14, Sensimetrics). Every subject heard the same music clips. This headphone model can attenuate scanner noise and has been widely used in previous MRI studies with auditory stimuli (Norman-Haignere et al., 2015). Scanning was performed using a 3.0T MRI scanner (TIM Trio; Siemens, Erlangen, Germany) equipped with a 32-channel head coil. For functional scanning, we scanned 68 interleaved axial slices with a thickness of 2.0mm without a gap using a T2*-weighted gradient echo multi-band echo-planar imaging (MB-EPI) sequence (Moeller et al., 2010) (repetition time (TR, aka. sampling interval) = 1,500ms, echo time (TE) = 30ms, flip angle (FA) = 62°, field of view (FOV) = $192 \times 192$mm$^2$, voxel size = $2 \times 2 \times 2$mm$^3$, multi-band factor = 4). A total of 410 volumes were obtained for each run.

**Preprocessing details.** We pre-process the *music genre neuroimaging dataset*[5] from Nakai et al. (2022) in a same fashion as Nakai et al. (2021), recited below. Motion correction is performed for each run using the Statistical Parametric Mapping toolbox (SPM8; Wellcome Trust Centre for Neuroimaging, London, UK; fil.ion.ucl.ac.uk/spm). All volumes are aligned to the first EPI image for each participant. For each clip, 2s of fade-in and fade-out were applied and the overall intensity was normalized. Low-frequency drift is removed using a median filter with a 240s window. To augment model fitting accuracy, the response for each voxel is normalized by subtracting the mean response and then scaling it to the unit variance. We use FreeSurfer (Dale et al., 1999) to identify the cortical surfaces from the anatomical data and register them with the voxels of the functional data. We use only cortical voxels as targets of the analysis for each participant. For each participant, we use the voxels identified in the cerebral cortex in the analysis (53,421 to 64,700 voxels per participant).

### A.1.2 MUSICLM

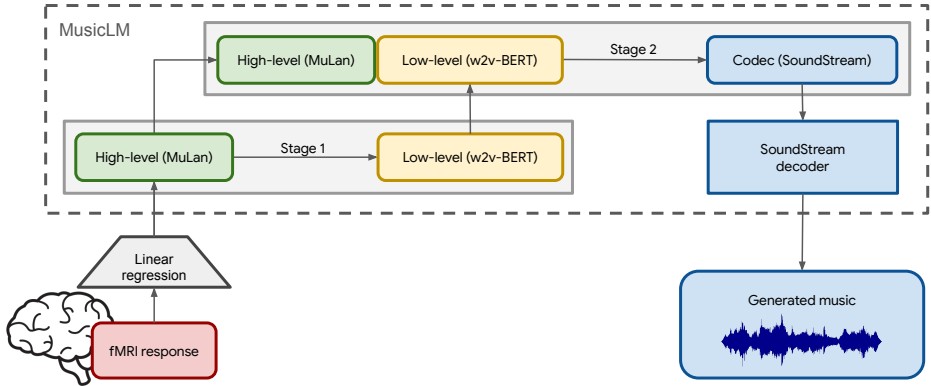

Figure A.1: Visual representation of MusicLM (Agostinelli et al., 2023) in the context for fMRI decoding. Rounded-rectangle elements denote embeddings/tokens; sharp corners models with parameters. The process begins with an fMRI response, captured from a test subject exposed to music. It is subsequently mapped to a 128-dimensional MuLan embedding via linear regression. The first stage of MusicLM then refines the MuLan embedding into low-level representation of w2v-BERT tokens with temporal information. The subsequent stage, informed by both the output of the previous stage and the MuLan embedding, generates tokens for the SoundStream audio codec. In the last step these are transformed into a waveform through a SoundStream decoder.

---

[5]Download link: openneuro.org/datasets/ds003720

Table A.2: Different embeddings and their properties: $d$ denotes the embedding dimensionality. $r$ is the number of embeddings we compute per 15s music clip, when we advance the given window with a step size of 1.5s. We also provide the original frequency of the embeddings as provided by the models. After averaging across the given time window for SoundStream and w2v-BERT embeddings, the frequency becomes 0.67 Hz for all models, given our step size of 1.5s. For a comparison, we also include properties of the fMRI data of subject 1 (for which $d$ are the dimensions of the top 6 most-correlated ROIs when predicting MuLan$^{\text{music}}$ embeddings).

| Embedding | $d$ | $r$ | Original freq. [Hz] | Window size [s] |
|---|---|---|---|---|
| SoundStream-avg | 128 | 10 | 50 | 1.5 |
| w2v-BERT-avg | 1024 | 7 | 25 | 5 |
| MuLan$^{\text{music}}$ | 128 | 4 | 0.67 | 10 |
| MuLan$^{\text{text}}$ | 128 | 1 | - | 15 |
| fMRI (top-6 ROIs) | 1460 | 10 | 0.67 | 1.5 |

### A.1.3 HYPERPARAMETER TUNING

We use a ridge regression to estimate our model parameters. Citing the *himalaya* documentation[6]: Let $\boldsymbol{X} \in \mathbb{R}^{n \times p}$ be a feature matrix with $n$ samples and $p$ features, $\boldsymbol{y} \in \mathbb{R}^n$ a target vector, and $\alpha > 0$ a fixed regularization hyperparameter. Ridge regression (Hoerl & Kennard, 1970) defines the weight vector $\boldsymbol{b}^* \in \mathbb{R}^p$ as:

$$\boldsymbol{b}^* = \arg\min_{\boldsymbol{b}} \|\boldsymbol{X}\boldsymbol{b} - \boldsymbol{y}\|_2^2 + \alpha\|\boldsymbol{b}\|_2^2. \tag{2}$$

The equation has a closed-form solution $\boldsymbol{b}^* = \boldsymbol{M}\boldsymbol{y}$, where $\boldsymbol{M} = (\boldsymbol{X}^\top \boldsymbol{X} + \alpha \boldsymbol{I}_p)^{-1} \boldsymbol{X}^\top \in \mathbb{R}^{p \times n}$.

To determine $\alpha$ we run 5-fold cross-validation on the training data. Note that there is one $\alpha$ parameter per regression target, i.e., 128 in our case when predicting MuLan embeddings. We inspect the performance on training and evaluation data around the chosen $\alpha$ vector "$\alpha$ (opt)" in Figure A.2.

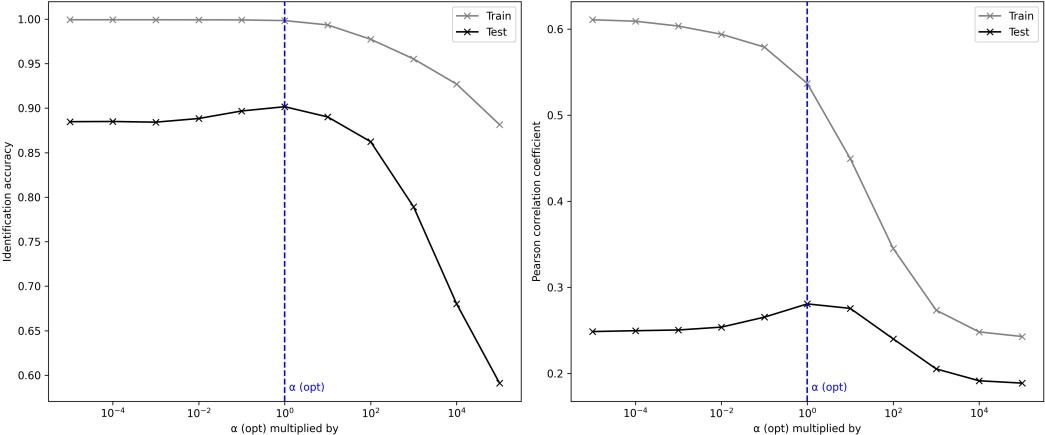

Figure A.2: Performance of the regressor when trained with $\alpha$ values in the neighborhood of the $\alpha$ that was determined to be optimal on the *training split* via cross-validation. The model starts to overfit with lower values of $\alpha$ (to the left) and underfits in the opposite direction. The plot is computed for a model predicting MuLan embeddings for subject 1.

### A.1.4 IDENTIFICATION ACCURACY METRIC

Assume there is a matrix of predicted embeddings $\boldsymbol{P} \in \mathbb{R}^{n \times d}$ and a matrix (of equal size) containing target embeddings $\boldsymbol{T}$. Let $\boldsymbol{C} \in \mathbb{R}^{n \times n}$ be computed from $\boldsymbol{P}$ and $\boldsymbol{T}$, specifically $\boldsymbol{C}_{i,j}$ is Pearson correlation coefficient between $i$-th row of $\boldsymbol{P}$ and $j$-th row of $\boldsymbol{T}$. The identification accuracy for the

---

[6]gallantlab.org/himalaya/models.html#ridge

$i$-th prediction is defined as:

$$\text{id acc}_i = \frac{1}{n-1} \sum_{j=1}^{n} \mathbb{1}\left[C_{i,i} > C_{i,j}\right].\tag{3}$$

The identification accuracy for all examples is simply the average:

$$\text{id acc} = \frac{1}{n} \sum_{i=1}^{n} \text{id acc}_i.\tag{4}$$

This score, ranging from 0 to 1 with 0.5 indicating performance equivalent to random chance, provides a quantified measure of how well an embedding was predicted in relation to other embeddings in the dataset.

An intuitive view on the identification accuracy is that a model with 86% such accuracy, on average, 14% of the candidates retrieved will score higher, i.e., have a higher correlation coefficient, than the correct candidate. In a dataset with 60 examples, the average rank of the correct music clip would be $0.14 \times 60 \approx 8$.

### A.1.5  Top-$n$ Class Agreement Metric

Below is the algorithm used to compute the top-$n$ class agreement. It is the intersection-over-union among the top-$n$ classes (of a certain group; see Section A.1.6) predicted by the LEAF (Zeghidour et al., 2021b) classifier operating on two music clips.

---

**Algorithm 1** Calculate top-$n$ class agreement

---

**Require:** $classesA$: ordered list of class labels for audio clip $A$
**Require:** $classesB$: ordered list of class labels for audio clip $B$
**Require:** $n$: number of top-$n$ classes to compare
 1: **function** TOPNCLASSAGREEMENT($classesA$, $classesB$, $n$)
 2:     $topA \leftarrow$ first $n$ elements of $classesA$
 3:     $topB \leftarrow$ first $n$ elements of $classesB$
 4:     $intersection \leftarrow$ set of common elements in $topA$ and $topB$
 5:     $union \leftarrow$ set of all unique elements in $topA$ and $topB$
 6:     **return** $\frac{\text{size of } intersection}{\text{size of } union}$
 7: **end function**

---

### A.1.6  AudioSet Class Groups

In our AudioSet evaluation metric we compute the overlap of the top-$n$ most likely classes (between reconstructed and original music) in three different groups. The groups and the $n$ choice are genres (top-2), instruments (top-2), and moods (top-1).

Below is a list of the AudioSet class names of each group:

Genres: *Pop music, Hip hop music, Rock music, Rhythm and blues, Soul music, Reggae, Country, Funk, Folk music, Middle Eastern music, Jazz, Disco, Classical music, Electronic music, Music of Latin America, Blues, Music for children, New-age music, Vocal music, Music of Africa, Christian music, Music of Asia, Ska, Traditional music, Independent music*

Instruments: *Plucked string instrument, Keyboard (musical), Percussion, Orchestra, Brass instrument, Bowed string instrument, Wind instrument, woodwind instrument, Harp, Choir, Bell, Harmonica, Accordion, Bagpipes, Didgeridoo, Shofar, Theremin, Singing bowl, Scratching (performance technique)*

Moods: *Happy music, Sad music, Tender music, Exciting music, Angry music, Scary music*

### A.1.7  Principal component analysis on the weight matrix of the encoding model

To further examine information representation in each voxel, we perform PCA on the weight matrix of encoding model of MuLan$^{\text{music}}$. For each of the top 600 voxels of predictive power (approximately 1% of all voxels, almost all in the auditory cortex), the weight matrix (voxels $\times$ 128) estimated by the MuLan encoding model

was extracted. The weight matrices (voxels $* 5 \times 128$) of the five participants were concatenated to obtain results for the whole group of participants. Dimensionality reduction of the weight matrices was performed using PCA on the feature direction of the concatenated weight matrices.

### A.1.8 VARIANCE PARTITIONING ANALYSIS

Although MuLan[music] and MuLan[text] predict similar voxels, they may represent different aspects of each voxel's activations. To explore this,, we combine MuLan[music] and MuLan[text] into a single model. We then examine their distinct contributions by mapping the unique variance explained by each feature onto the cortex, utilizing banded ridge regression (la Tour et al., 2022). Figure A.3 shows that MuLan[text] exhibits less unique explained variance compared to MuLan[music]. This observation differs notably from trends in visual neuroscience (Takagi & Nishimoto, 2022), where textual representations have distinct prediction performance. This discrepancy may indicate a unique aspect of musical processing. . In summary, although MuLan[text] predicts many areas within the auditory cortex, most of the information contributing to this prediction is contained in MuLan[music].

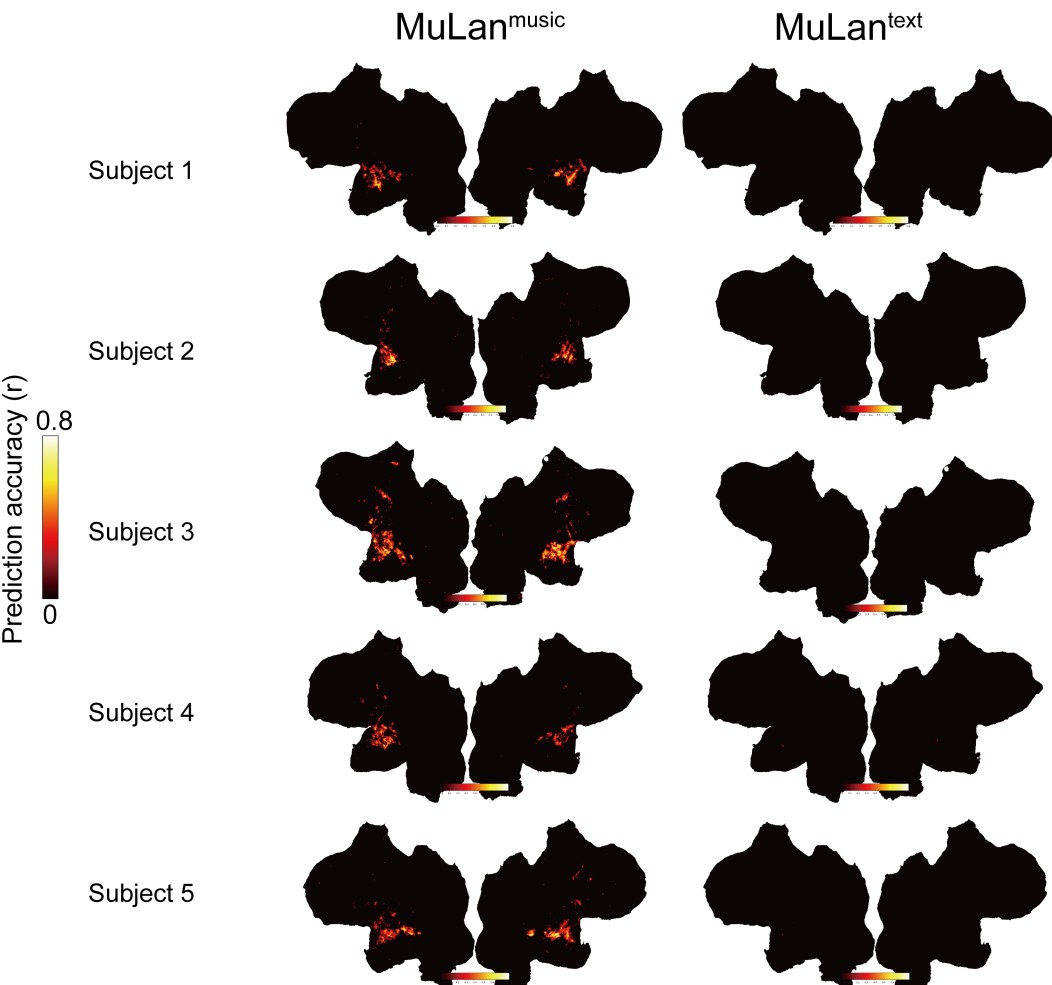

Figure A.3: All subject results for comparing unique variance explained by MuLan[music] and MuLan[text].

### A.1.9 OTHER AUDIO-DERIVED EMBEDDINGS

In addition to MuLan[music] and w2v-BERT, we develop separate encoding models using three different approaches - wav2vec 2.0 (Baevski et al., 2020), HuBERT (Hsu et al., 2021), and melspectrogram. Both wav2vec 2.0 and HuBERT are transformer-based encoders trained through self-supervised learning on 960 hours of LibriSpeech. For our study, we utilized the hubert-large-ls960-ft and wav2vec2-large-960h models available on Huggingface. We employ the outputs from the last hidden layer of both architectures. The melspectrogram was computed using Librosa. All other settings remained consistent with those used in MuLan[music].

## A.2 ADDITIONAL RESULTS OF THE DECODING

The two ways of reconstructing music which we compare are retrieval from FMA and generation with MusicLM. Figure A.4 contains qualitative results for subject 1, comparing the original stimulus to music retrieved from FMA based on a predicted MuLan embedding and the music clips sampled from MusicLM. Note that sampling multiple clips and examining their differences is one way of qualitatively determining which information MusicLM adds to what the MuLan embedding contains. We find that both retrieved and generated constructions are semantically similar to the original stimulus, e.g., with respect to genre, vocal style, overall mood. The temporal structure of the stimulus is often not preserved in the reconstruction. There are also failure cases in which the reconstruction is from an entirely different genre.

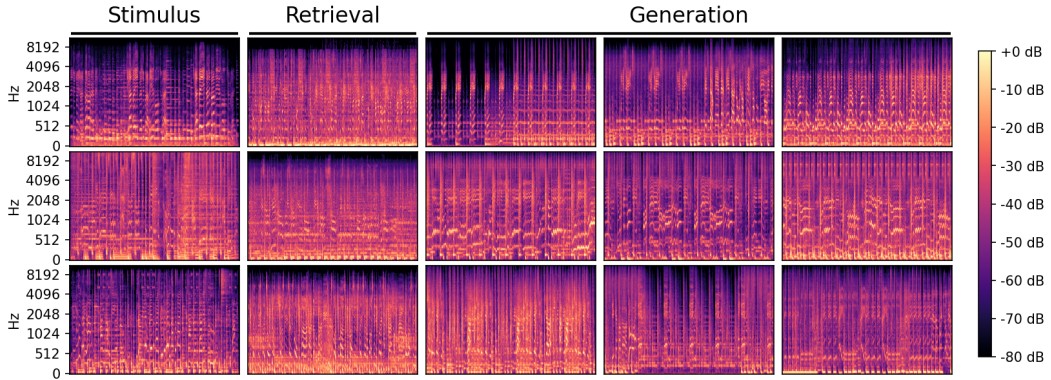

Figure A.4: Spectrograms of different music clips: The left-most column contains the stimulus which subjects were exposed to. To the right is the music retrieved from FMA and three clips sampled from MusicLM. Both generation and retrieval are done via MuLan embeddings. It is visually perceptible that spectrograms in the same row resemble similarities. Audio examples (randomly sampled, one per genre) can be found at f2mu.github.io#ret-vs-gen

In Figure A.5 we perform a comparison of retrieval and generation across the five different subjects. The main finding is that qualitatively, the reconstruction is overall of consistent quality across all five subjects. This is not necessarily a given, when dealing with fMRI data, because of differences in subjective experiences, and suggests our method is robust.

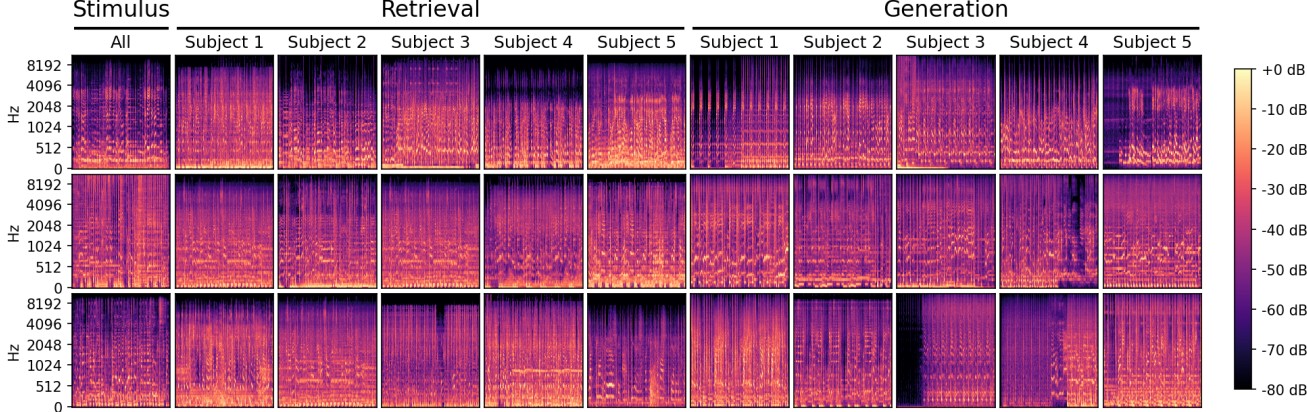

Figure A.5: The comparison shows the spectrograms of music retrieved or generated for different test subjects, who were all exposed to the same stimuli. The predicted embedding is MuLan. Audio examples (randomly sampled, one per genre) can be found at f2mu.github.io#all-subjects

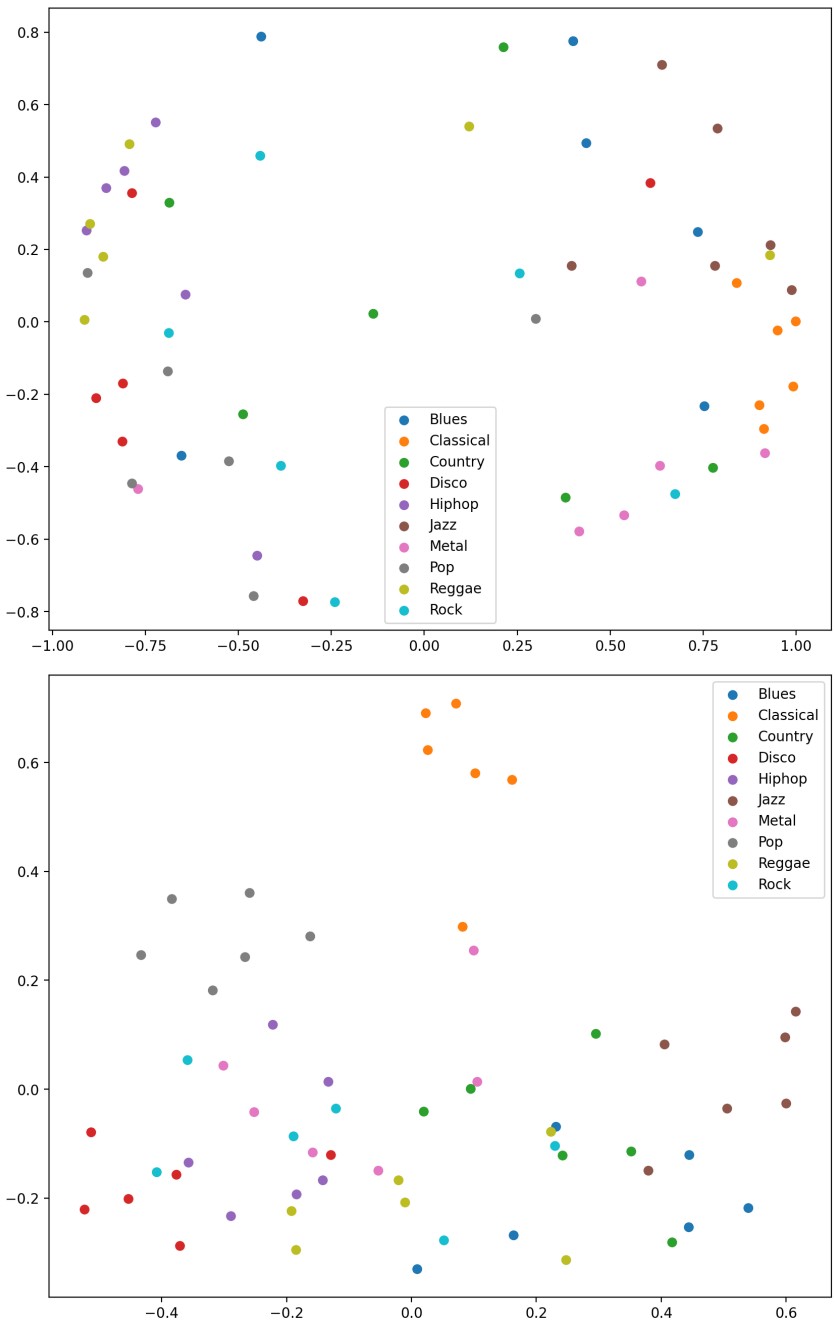

Figure A.6: Juxtaposition of PCA-dimensionality-reduced MuLan embeddings. Predicted embeddings are at the top and the ground truth at the bottom; all on the 60 evaluation data points with 10 genres. The resulting plot shows a non-random formation of clusters, for both ground truth and prediction, most prominently observable for the genres Classical, Hiphop, as well as Jazz. Note that the MuLan embedding space captures many more musical concepts than just genres and clusters of other kinds may for, such as vocal intensity, mood, or even musical complexity.

## A.3 ADDITIONAL RESULTS OF ENCODING

We present additional results for all subjects in Figures A.7, A.8, and A.9 for Figures 2, 4, and 5. They show that our results are robust across subjects.

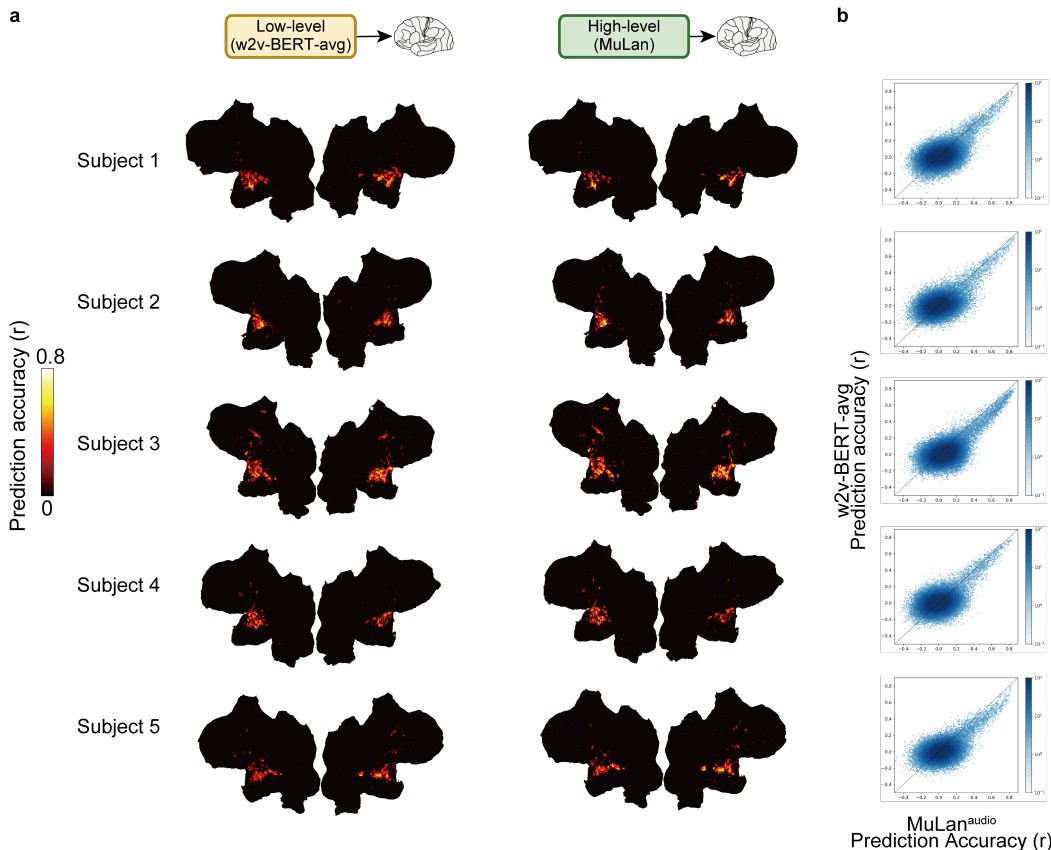

Figure A.7: **a** All subject results for comparing prediction performances between different audio-derived embeddings: MuLan[music] and w2v-BERT-avg. **b** Density plot of the MuLan[music] (x-axis) versus w2v-BERT-avg (y-axis) model prediction accuracy. Darker colors indicate a higher number of voxels in the corresponding bin.

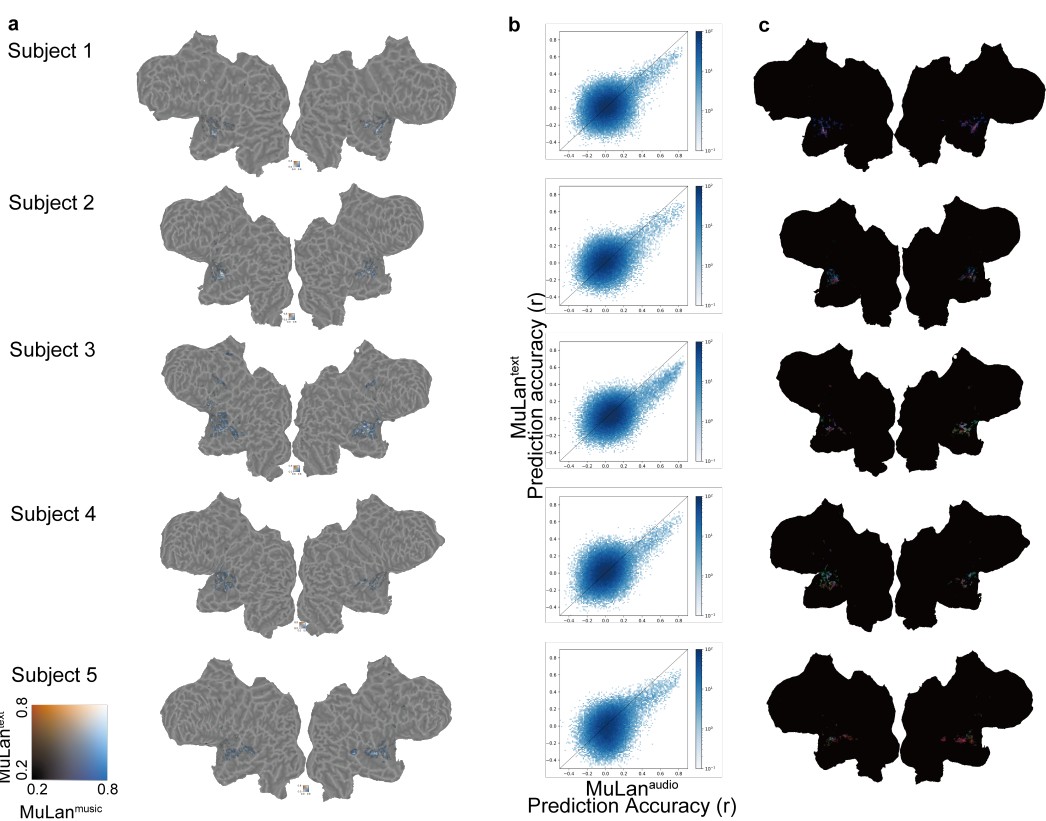

Figure A.8: **a** All subject results for comparing prediction performances between MuLan[music] and MuLan[text]. **b** Density plot of the MuLan[music] (x-axis) versus MuLan[text] (y-axis) model prediction accuracy. **c** Principal components analysis.

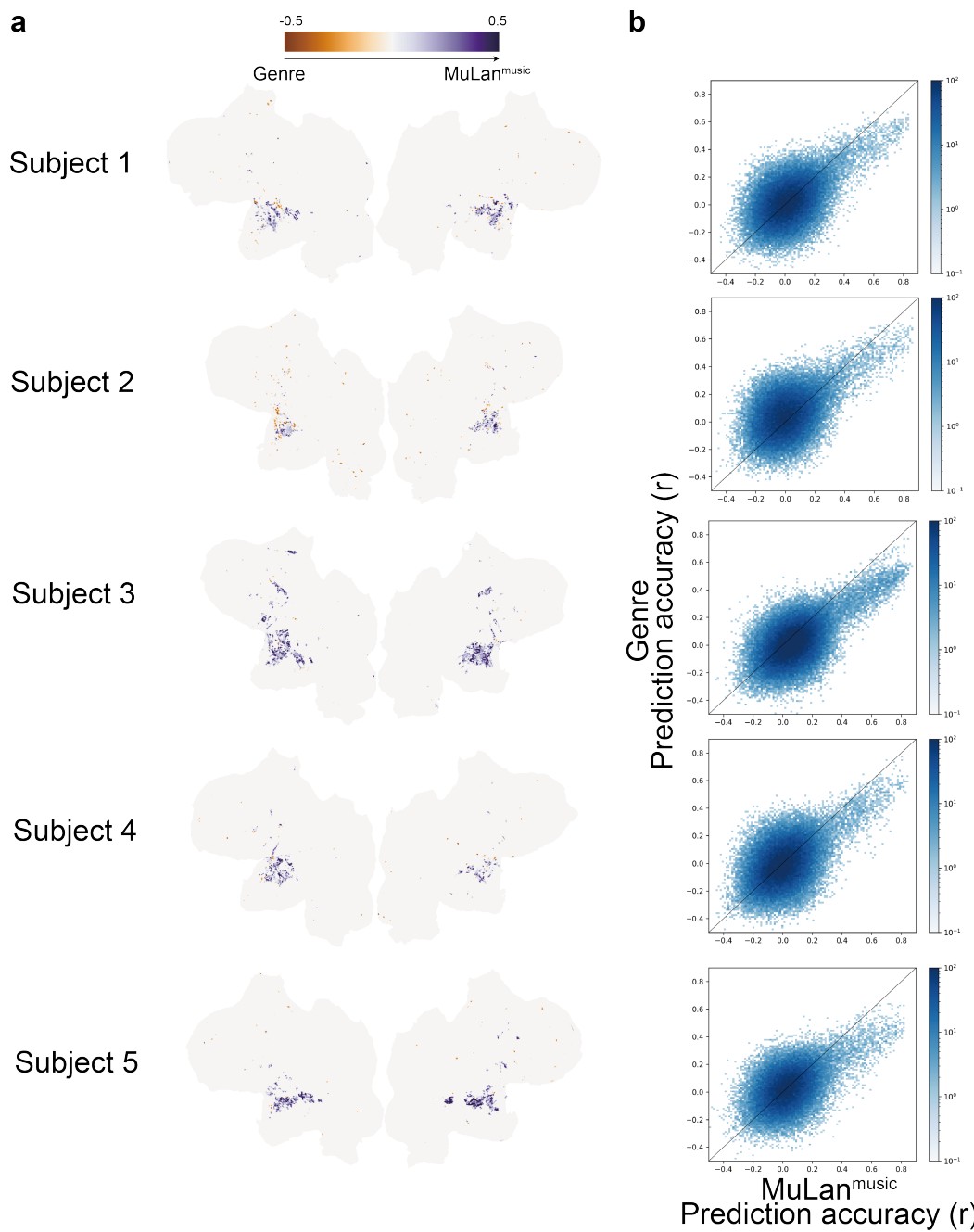

Figure A.9: **a** All subject results for comparing prediction performances between MuLan[music] and GTZAN genre label. **b** Density plot of the MuLan[music] (x-axis) versus genre (y-axis) model prediction accuracy. Many voxels are better predicted by MuLan[music] than genre.

## A.4 TEXT CAPTION DATASET

We release a text caption dataset[7] for the 540 GTZAN music clips (15s crop) for which fMRI data as recorded. The captions were collected with the online freelance platform coconala.com by human raters, all of which are music professionals (musician, teacher, composer). The instruction given to the raters is:

*We have numerous 15-second clips for which we'd like you to provide a written description of about four sentences in Japanese (or English). By written description, we mean something that includes an explanation or impression of the music piece, as demonstrated in the following example: "This is a Drum & Bass track. It features high-speed scratching from a turntable and includes sampled screams. You can hear a sinister tune being played by the synthesizer. The rhythmic backdrop is composed of fast electronic drum beats. This track seems like it could be used as a soundtrack for a car racing game."*

Below are ten example captions; one per genre. A table with audio and captions side by side is at f2mu.github.io#gtzan-caps

blues.00017: *It is lazy blues with a laid-back tempo and relaxed atmosphere. The band structure is simple, with the background rhythm punctuated by bass and guitar cutting. The impressive phrasing of the lead guitar gives the piece a nostalgic impression.*

classical.00008: *Several violins play the melody. The melody is simple and almost unison, but it moves between minor and major keys and changes expression from one to the other.*

country.00012: *This is a classic country song. You can hear clear singing and crisp acoustic guitar cutting. The wood bass provides a solid groove with a two-beat rhythm. This is country music at its best. Ideal for nature scenes and homely atmospheres.*

disco.00004: *This music piece has a disco sound. Vocals and chorus create extended harmonies. The synthesiser creates catchy melodies, while the drumming beats rhythmically. Effective tambourine sounds accentuate the rhythms and add further dynamism. This music is perfect for dance parties, club floors and other scenes of dancing and fun.*

hiphop.00014: *This is a rap-rock piece with a lot of energy. The distorted guitars are impressive and provide an energetic sound. The bass is an eight beat, creating a dynamic groove. The drums provide the backbone of the rhythm section with their powerful hi-hats. The vocal and chorus interaction conveys tension and passion and draws the audience in.*

jazz.00040: *This is medium-tempo old jazz with female vocals. The band is a small band similar to a Dixie Jazz formation, including clarinet, trumpet and trombone. The vocal harmonies are supported by a piano and brass ensemble on a four beat with drums and bass.*

metal.00026: *This is a metal instrumental piece with technical guitar solos and distortion effects. The heavy, powerful bass creates a sense of speed, and the snare, bass and guitar create a sense of unity in unison at the end. It is full of over-the-top playing techniques and intense energy.*

pop.00032: *Passionate pops piece with clear sound and female vocals. The synth accompaniment spreads out pleasantly and the tight bass grooves along. The beat-oriented drums drive the rhythm, creating a strong and lively feeling. Can be used as background music in cafés and lounges to create a relaxed atmosphere.*

reggae.00013: *This reggae piece combines smooth, melodic vocals with a clear, high-pitched chorus. The bass is swingy and supports the rhythm, while whistles and samplers of life sounds can be heard. It is perfect for relaxing situations, such as reading in a laid-back café or strolling around town.*

rock.00032: *This rock piece is characterised by its extended vocals. The guitar plays scenically, while the bass enhances the melody with rhythmic fills. The drums add dynamic rhythms to the whole piece. This music is ideal for scenes with a sense of expansiveness and freedom, such as mountainous terrain with spectacular natural scenery or driving scenes on the open road.*

---

[7]f2mu.github.io/caption-dataset.csv

