# OpenReview forum: "Brain2Music: Reconstructing Music from Human Brain Activity"
_ICLR.cc/2024/Conference — Submitted to ICLR 2024_

### Official Review · Reviewer_WEzt · 2023-10-25

**Soundness:** 2 fair
**Presentation:** 4 excellent
**Contribution:** 2 fair
**Rating:** 3
**Confidence:** 5

**Summary:**

This paper proposed a framework named BRAIN2MUSIC that maps fMRI data to music waveforms. BRAIN2MUSIC is based on pre-trained models MuLan and MusicLM. MuLan is a text/music embedding model with two encoders. MusicLM is a conditional music generation model. The authors align the fMRI representation with MuLan embeddings and use the embedding as conditions for music generation. The mapping between MuLan representation and fMRI is done by applying linear regression. In addition, the authors utilized a publicly available dataset music genre neuroimaging dataset to verify the performance of the proposed BRAIN2MUSIC.

**Strengths:**

This paper is well organized with a clear presentation.

**Weaknesses:**

* a) The main drawback of this work is that the conclusion drawn is not persuasive by adopting powerful pre-trained models and the limited data size of the dataset utilized in this work. Mulan is a music text embedding model that's pre-trained on a large amount of music/text data. Similarly, MusicLM is a powerful conditional music generation model pre-trained on large amounts of data. Say we randomly sample embeddings from the hidden state space of Mulan and use them as a condition to guide MusicLM; we get a piece of 'meaningful' music. By meaningful, I mean it sounds like music, not noise. Now, linearly mapping fMRI to the embedding space of Mulan to generate music does not necessarily show the correspondence between fMRI and music but the powerful representation ability of Mulan and the generation ability of MusicLM. I also carefully listened to the demos given. This confirmed my thought that the generated or reconstructed music sounds like real music but is not very similar to the stimulus music. This is also verified by the low correlation results demonstrated in Table 1. It would also be interesting to see the correlation result of only using the mean vector or random vectors to reconstruct the music.

* b) No visualization results of the embeddings from fMRI to Mulan are given. Ideally, the embeddings $\hat{T}$ on the test set should show a clustering effect.

* c) Given the small size of the dataset, a K-fold evaluation should be adopted rather than a fixed test set with only 60 data points.

* d) No comparisons are conducted w.r.t other works. Especially seq2seq approaches, meaning directly predicting music waveform using fMRI. Thus, it is hard to evaluate the solidness of this work.

**Questions:**

See weakness.

---

> ### Author Response · Authors · 2023-11-16
> **First Response to Reviewer WEzt**
>
> We thank you very much for the review and would like to address the weaknesses pointed out:
>
> a) While we acknowledge in Section 5 (Limitations) that our system's representation power is limited by MuLan, we emphasize that our quantitative and qualitative results demonstrate that our system's performance significantly outperforms chance (randomly sampling from the MuLan embedding space). While the use of MusicLM contributes to the generation of pleasant-sounding music, the results presented in Table 1 and Figure 1 would not surpass chance levels without the extraction of meaningful information from the input signal (fMRI recordings).
>
> b) Thanks for this suggestion. We ran a new analysis in which we take all MuLan embeddings predicted on the validation data and reduce their dimensionality from 128 down to 2 using a PCA. The resulting plot (with colors indicating genres) can now be found in [Figure A.6](https://github.com/f2mu/f2mu.github.io/blob/main/rebuttal-figures/fig-a6.png?raw=true) in the updated paper. We observe that the predictions form noisy clusters, most prominently for the genres _Classical_, _Hiphop_, and _Jazz_. To get a better feel for what to expect, we also PCA-dimensionality-reduced the ground truth MuLan embeddings (computed directly from the GTZAN eval data) and plotted them in juxtaposition.
>
> c) During the fMRI recording, our test-set is repeated several times to enhance SNR. This is a common approach in fMRI encoding/decoding analyses. It’s not suitable for a k-fold CV, because the test examples are available with lower SNR. Apart from that, they are recorded in a separate run (i.e., temporally separated) from the train examples to be as certain as possible that there is no leakage (details are in Nakai et al. (2022)).
>
> d) In terms of comparison with other models, in our response to Reviewer QVSZ, we report the new results of constructing encoding models using different models. A detailed comparison of the prediction performance between different models and different brain regions yielded interesting insights. Particularly in terms of prediction performance, MuLan was found to be the most effective in predicting the auditory cortex during listening music.
>
> Furthermore, to the best of our knowledge, there are no published works that have performed music reconstruction using seq2seq in fMRI. Therefore, we believe our study is the first to establish a correspondence between fMRI brain activity for music and DNN features of music.

---

### Official Review · Reviewer_fWGs · 2023-10-31

**Soundness:** 2 fair
**Presentation:** 3 good
**Contribution:** 2 fair
**Rating:** 3
**Confidence:** 4

**Summary:**

The authors apply decoding and encoding analyses to fMRI responses to music using recently developed Transformer-based music generation models.

For decoding analyses, they learn a linear weighting of voxel activity that best maps onto the embeddings from different components of the MusicLM model. They then either select a clip that best matched the predicted embedding from a corpus (FMA) or they use the generation capacity of the model to generate a waveform. They show above chance ability to decode the model features, with the highest identification accuracy for the music embedding of MuLan. They report that they can better recover semantic properties, such as genres, instruments, and moods using the generative approach.

For the encoding analyses, they learn a linear map from different features of the MusicLM model to voxel data. They report that voxels are similarly well predicted by the w2v-BERT-avg and MuLan. They show that the encoding model predictions are better for the music variant of MuLan compared with the text variant consistent with better decoding. They also perform PCA on the weights from the learned encoding model and plot the stimulus and voxel embeddings.

**Strengths:**

Exploring whether modern transformer-based music models can improve encoding and decoding models in the brain is a good idea. There has been a lot of progress in this area and would be interesting to know whether these models learn representations that mirror the brain to any extent.

Leveraging the generative aspects of these models for the purpose of decoding is also potentially interesting. There could be methodological value and future scientific insights gained from developing improved decoding models for music.

**Weaknesses:**

The analyses are fairly preliminary and there are currently no clear neuroscience insights.

There are no comparisons against standard acoustic models used in the auditory neuroscience literature. For example, it is unclear whether the decoding model performs better at identification compared with the standard spectrotemporal modulation transfer model tested in Zakai. There is also no comparison against other DNN audio models such as wav2vec2.0 or HuBERT which have shown promising prediction accuracy in auditory cortex (the relation between w2v-BERT-avg and these prior models is unclear).

There are no perceptual experiments done to evaluate the quality of the reconstructions.

There is no serious investigation of how encoding and decoding results might vary across the auditory hierarchy.

There is no investigation of how performance might vary across different layers of the network in the case of the encoding models.

For encoding models there are no attempts to estimate the unique contribution of different models by comparing the performance of individual models against combined models. This is important as the features from different models are highly correlated. Thus a text-based model might predict auditory responses due to correlated features rather than a genuine response to text. As a consequence, the scatter plots showing correlated predictions is not surprising or particularly informative.

The statistical approach used to compute p-values does not seem appropriate, since it assumes the samples are independent and Gaussian distributed. They could use a permutation test across stimuli as an alternative.

Some of evaluation metrics were unclear to me (see questions below).

**Questions:**

I found some of metrics difficult to understand. For Figure 1A, isn’t the identification accuracy based on the latent embeddings? How is this based on the reconstructions? Can you spell out exactly what was done to compute this figure.

For Figure 1B and 1C, how were the genre, instrument, and mood labels determined? Please also give the equations for how overlap was computed.

---

> ### Author Response · Authors · 2023-11-16
> **Response to Reviewer fWGs (1/2)**
>
> We thank the reviewer very much for their review! Below we're addressing weaknesses and questions separately.
>
> **Rebuttal to weaknesses pointed out:**
>
> Motivated by your feedback, we have conducted new experiments and revised our manuscript following the insightful comments we received. This process uncovered several new insights, particularly in identifying previously unknown feature selectivity across hierarchy within the auditory cortex. We believe these revisions have significantly enriched the neuroscience aspect of our work. These additions have been included as new figures in the revision.
>
> > The analyses are fairly preliminary and there are currently no clear neuroscience insights.
>
> Our study is the first fMRI study to explore the relationship between the brain activity during music listening and DNNs for music generation. Given that music is very complex, the significance of finding models that accurately describe brain activity during musical engagement is substantial. Therefore, we believe that the findings of our study are of great importance.
>
> However, the reviewer's comments have been extremely beneficial in making our research more intriguing. In response to the comments, we have conducted several additional analyses as follows.
>
> > There are no comparisons against standard acoustic models used in the auditory neuroscience literature. [...]
>
> Following the comments, we constructed encoding models using non-DNN features (Mel-spectrogram), which are thought to capture lower-level music features, and different DNN features (Hubert and wav2vec2.0). Additionally, to address subsequent questions, we compared the prediction performance of each layer of the auditory cortex. Our findings are summarized as follows (please see the newly added [Figure 3](https://raw.githubusercontent.com/f2mu/f2mu.github.io/main/rebuttal-figures/fig-3.png) in the revised manuscript):
>
> 1. MuLan demonstrated superior overall performance compared to the other models. Since it is an integral component of MusicLM, this may indicate the effectiveness of music generation models in exploring brain activity related to music.
> 2. As anticipated, the prediction accuracy of the mel spectrogram decreased with the increasing hierarchy of the auditory cortex.
> 3. However, strikingly, DNN features more accurately predicted the Lbelt/Pbelt regions (also known as A2/A3; early layers within the auditory hierarchy) than A1 (the earliest, primary auditory cortex), yet demonstrated lower performance in A4/A5 (association, higher auditory cortex). This trend has not been observed in other auditory neuroscience studies that focus on general sounds or speech, suggesting that it might be a phenomenon unique to music motivating further exploration.
> 4. The observed decline in the prediction performance of mel spectrogram with ascending hierarchy suggests that the results for DNN features might reflect inherent differences in the features themselves, rather than mere differences in signal-to-noise ratio (SNR) across different ROIs.
>
> In summary, thanks to the reviewer, we have gained new and interesting insights into the neural representation of music in neuroscience and machine learning models.
>
> > There are no perceptual experiments done to evaluate the quality of the reconstructions.
>
> During the short rebuttal period it is not feasible for us to undertake the requested rater study. We recognize this as a valuable addition for future work.
>
> > There is no serious investigation of how encoding and decoding results might vary across the auditory hierarchy.
>
> This is also an important point. As mentioned above, we now compared the prediction performance of each layer of the auditory cortex and found the results as described.
>
> > There is no investigation of how performance might vary across different layers of the network in the case of the encoding models.
>
> Indeed, we believe this is interesting to explore in addition, but due to time constraints during the rebuttal period, we would like to address it as a future task.

---

> > ### Author Response · Authors · 2023-11-16
> > **Response to Reviewer fWGs (2/2)**
> >
> > > For encoding models there are no attempts to estimate the unique contribution of different models by comparing the performance of individual models against combined models. [...]
> >
> > Thank you very much for your feedback. We acknowledge that our initial analysis already demonstrates that both MuLanText and MuLanMusic effectively predict similar voxels, with MuLanMusic showing superior performance. However, we agree with your suggestion that a further analysis could elucidate how these two representations differ in their predictive capabilities. Accordingly, we conducted an investigation using banded ridge regression (la Tour et al., 2022) to assess the unique variance of these features. Our findings are summarized below (please see new supplementary [Figure A.3](https://github.com/f2mu/f2mu.github.io/blob/main/rebuttal-figures/fig-a3.png?raw=true) in the revised manuscript):
> >
> > 1. MuLanText lacks unique explained variance when compared to MuLanMusic.
> > 2. This observation is notably different from trends in visual neuroscience (e.g. Takagi and Nishimoto, CVPR 2023), where text embeddings have distinct prediction performance. This might suggest a phenomenon unique to musical processing.
> > 3. Our additional analysis is complementary to the initial analysis: although MuLanText predicts many areas within the auditory cortex, most of the information contributing to this prediction is contained in MuLanMusic.
> >
> > In conclusion, the insightful suggestions from the reviewer have enabled us to uncover new perspectives in neuroscience research.
> >
> > > The statistical approach used to compute p-values does not seem appropriate, since it assumes the samples are independent and Gaussian distributed. They could use a permutation test across stimuli as an alternative.
> >
> > Thank you for the important point. We changed to a more conservative p-value using a permutation test for the encoding model and all figures have been replaced. We confirmed that the auditory cortex was still central to the predictions, and there were no changes to any of the interpretations.
> >
> > **Responses to questions:**
> >
> > Figure 1a: The identification accuracy is obtained by computing the embedding (MuLan or w2v-BERT-avg) of the reconstructed music.
> >
> > The idea is to evaluate end to end (in the chain: music stimulus → fMRI scans → MusicLM→ music reconstruction) by comparing the similarity of music stimulus and music reconstruction in two embedding spaces. For that we compute the embeddings of the music stimulus and those of the reconstruction and determine the identification accuracy based on them. We've updated the respective caption in the paper to make this clearer.
> >
> > Figure 1b: The genre, instrument, and mood labels were computed using the LEAF (Zeghidour et al., 2021b) classifier. This is mentioned in Section 3.3 in the 'Evaluation metrics'. To make the computation clearer we've added [Section A1.5](https://github.com/f2mu/f2mu.github.io/blob/main/rebuttal-figures/algo-1.png?raw=true) to the appendix which explicitly states the algorithm used.
> >
> > Figure 1c: The genres here are the GTZAN labels. (Why not use them for Figure 1b as well? Because they would not be available for the music reconstruction, whereas the LEAF model can be applied to any music clip.) Computation of the identification accuracy follows the same protocol as Figure 1a, just drilled down into the genres.

---

> ### Comment · Reviewer_fWGs · 2023-12-02
> **Summary of review**
>
> I thank the authors for responding to my comments.
>
> My overall take remains the same in that there are no substantial new scientific insights. The authors have added a new figure that shows better performance for DNN models compared with a mel spectrogram. This is not surprising given prior work (e.g., see Kell et al., 2018, A Task-Optimized Neural Network Replicates Human Auditory Behavior, Predicts Brain Responses, and Reveals a Cortical Processing Hierarchy). They seem to make a big deal out of lower performance in A4/A5, but this could just be explained by lower reliability and it is not particularly surprising that as you move out of the auditory cortex these models would perform more poorly. There also are not any statistics to support the claims made for this section.
>
> The appendix figure A.3 is interesting in providing a way to show the weak contribution of semantics, but an appendix figure is not going to change the overall contribution of a paper.
>
> I am therefore not going to change my score.

---

### Official Review · Reviewer_QVSZ · 2023-10-31

**Soundness:** 3 good
**Presentation:** 3 good
**Contribution:** 3 good
**Rating:** 8
**Confidence:** 3

**Summary:**

This paper proposes a Brain to Music pipeline to reconstruct music from fMRI data. More specifically, this pipeline contains two key components: (1) MuLan, a text/music embedding model, and (2) MusicLM, a conditional music generation model. The pipeline first uses fMRI recordings to predict music embeddings by a regularized linear regression; then, it applies the predicted music embeddings as conditions to MusicLM, where the MusicLM could recover or generate the corresponding music. In the experiments, this paper starts from a decoding task by quantitatively evaluating the music reconstruction; then, it illustrates the difference between text-derived and music-derived embeddings by designing an encoding task to predict fMRI recordings. Finally, this paper explores the generalization ability of the proposed pipeline.

**Strengths:**

* An interesting and novel topic.

* A clear and detailed writing of the related works and methods.

* A comprehensive experimental section. The authors discuss the proposed pipeline from both the decoding and encoding perspectives, which makes the role of involved components precise.

* A good discussion of the current limitations, e.g., the temporal sampling rate of fMRI may be too slow to collect high-frequency information.

**Weaknesses:**

* According to your [demos](https://f2mu.github.io), the presented examples are nearly music clips with a strong rhythm, which may be easy to reconstruct from fMRI. Could you give some instances where the music clips come from a symphony (with a weak rhythm)?

**Questions:**

* As the authors mentioned in section 5, the relatively high TR of fMRI is a limitation. Have you explored the retrieval/reconstruction performance of music clips with different frequencies?

* Are the fMRI recordings able to encode some complex music without a precise rhythm, like a symphony?

---

> ### Author Response · Authors · 2023-11-16
> **First Response to Reviewer QVSZ**
>
> We thank the reviewer very much for your review! Below we respond to the weaknesses and questions.
>
> Reconstruction of music without precise rhythm, e.g., a symphony: Out of the 10 genres that GTZAN contains, one is classical. A non-cherry-picked example from it is in the supplementary material in sections _Comparison of Retrieval and Generation_ and _Comparison Across Subjects_ in the second row. Generally speaking, information about the exact timing of beats is very unlikely to be contained in fMRI scans due to their low temporal resolution. We do see, however, that a non-existent beat in the stimulus also leads to no beat being in the reconstruction.
>
> fMRI sampling rate: We have not explored the reconstruction of music based on fMRI scans obtained at a higher sampling rate. Exploring this is interesting future work, perhaps alongside the use of EEG, for example. We thank the reviewer for this suggestion.

---

### Author Response · Authors · 2023-11-16
**First Summary Reply**

We thank the reviewers for their valuable comments. The reviewers remarked that our work is supported by a comprehensive experimental section (QVSZ), well organized with a clear presentation (WEzt), and detailed writing (QVSZ). The area we focus on is interesting and novel (QVSZ) and investigating in such a topic is a good idea which can provide methodological value and future scientific insights (fWGs).

We believe that our work provides a solid foundation for future scientific advancements, while acknowledging the limitations inherent in addressing novel scientific questions within the neuroscience domain. Below we address the concerns and questions raised by the reviewers.

---

### Meta-Review · Area_Chair_fvz6 · 2023-12-19

**Metareview:**

The submission puts forward a new task, that of reconstructing audio from neural recordings, particularly from fMRI. In a sense this is a new task, but in another, it is a well-understood domain with a fairly standard decoding task just using novel stimuli. This not a negative in itself, but, at several points in the author rebuttals serious reviewer concerns are addressed by noting “limitations inherent in addressing novel scientific questions within the neuroscience domain”. This manuscript would be far stronger had it followed the general approach and analysis used in other related work, even if the stimuli are different.

As the reviewers point out the statistical analysis performed is lacking. There are standard methods like permutation tests which are routinely used for such work. The reviewers point out that it is hard to believe, understand, and contextualize the answers without a more thorough analysis. The authors in their responses did add some extra analysis where they use other features which could potentially explain their decoding. But, even the analysis of those results suffers from the same problems, for example, “Figure 3 shows MuLanmusic outperformed the other models overall.” As the reviewers point out, it is unlikely given the error bars that such statements would pass a statistical test making the conclusions being drawn difficult to accept.

I encourage the authors to rework their analysis toward something that is more standard in this domain, from which they will be able to draw conclusions that are more likely to be accepted.

**Justification For Why Not Higher Score:**

Given the analysis provided, the conclusions that are drawn are not justified.

**Justification For Why Not Lower Score:**

N/A

---

### Decision · Program_Chairs · 2024-01-16

Reject